# The “K-Sign”—A Novel CT Finding Suggestive before the Appearance of Pancreatic Cancer

**DOI:** 10.3390/cancers13164222

**Published:** 2021-08-22

**Authors:** Yuko Kobashi, Masateru Uchiyama, Junichi Matsui

**Affiliations:** 1Department of Radiology, Tokyo Dental College Ichikawa General Hospital, Chiba 272-8513, Japan; wbtr8135@yahoo.co.jp; 2Department of Cardiovascular Surgery, Teikyo University, Tokyo 173-0003, Japan; mautiya@yahoo.co.jp; 3Department of Surgery, Tokyo Dental College Ichikawa General Hospital, Chiba 272-8513, Japan

**Keywords:** pancreatic invasive ductal adenocarcinoma, K-sign, the early diagnosis, computed tomography

## Abstract

**Simple Summary:**

This study investigated pancreatic morphological abnormalities leading to the early diagnosis of pancreatic ductal adenocarcinoma with computed tomography (CT) imaging. The “K-sign” detected in this study is an important CT finding as a pancreatic morphological abnormality that may be used as a reliable predictor of pancreatic cancer development.

**Abstract:**

Pancreatic invasive ductal adenocarcinoma (PDAC) has a poor prognosis, and the detection of PDAC during the early stage is thought to improve prognosis. In this study, we retrospectively investigated pancreatic morphological abnormalities that lead to the early diagnosis of PDAC with computed tomography (CT) imaging. In total, 41 out of 308 patients diagnosed with pancreatic cancer between 2011 and 2017 in our institution were enrolled. As a control group for the group with pancreatic cancer, 4277 patients without pancreato-biliary diseases were enrolled. We retrospectively reviewed and analyzed the clinical data including patient characteristics, the clinical course and preoperative CT imaging with pancreatic morphological features. Out of 41 patients, 24 patients (58.5%) showed local K-shaped constriction of the pancreatic parenchyma “K-sign” on preoperative CT images. Eight patients (19.5%) showed localized fatty change. Out of 4277 control patients, seven patients (0.16%) showed K-sign. “K-sign” may be used for the early diagnosis of PDAC by CT imaging.

## 1. Introduction

Each year, the National Cancer Center estimates the numbers of new cancer cases and deaths that will occur in Japan in the current year and collects the most recent data on cancer incidence, mortality, and survival. Approximately 40 years since 1981, when cancer (malignant neoplasm) became the leading cause of death in Japan, the number of patients dying from cancer continues to increase year by year. In 2019, 376,425 patients died from cancer, which accounted for approximately 30% of the country’s total deaths [1]. Additionally, approximately 100,000 new cancer cases have occurred in Japan. Statistically, the probability of cancer development in a lifetime is 62% in men and 47% in women, and that of cancer death is 25% in men and 16% in women [1]. In other words, one in two Japanese people will develop cancer and one in three Japanese people will die of cancer. In particular, pancreatic cancer was the fourth common malignancy followed by lung cancer, colon cancer, rectal cancer, and stomach cancer, which accounted for approximately 36,000 new cases in 2019 [1]. The five-year relative survival rate for patients with PC between 2009 and 2011 was 8.9% in men and 8.1% in women, which was the lowest of all cancers [2]. Furthermore, according to data from the Pancreatic Cancer Registry Report 2007, the median survival times with PC were 10 months in all patients, 12 months in resectable cases, and four months in unresectable cases, and five-year survival rates were 11.6%, 14.5%, and 0.3%, respectively [3].

Due to its poor prognosis and the end-stage at which it is typical to diagnosis early detection of PC, it is a fearful disease of which mortality almost parallels its incidence. However, pancreatic cancer is by no means an incurable cancer. Certainly, pancreatic cancer is hard to diagnose by subjective or clinical symptoms immediately. In general, many examinations such as abdominal ultrasonography, computed tomography (CT), magnetic resonance imaging, magnetic resonance cholangiopancreatography, tumor marker measurement, endoscopic ultrasonography, endoscopic retrograde cholangiopancreatography, and positron emission tomography are needed to be performed to make a definitive diagnosis of pancreatic cancer. Accurate diagnosis of pancreatic cancer by various examinations should help early detection of pancreatic cancer and improve the prognosis for patients with pancreatic cancer. Indeed, the Pancreatic Cancer Registry Report 2007, conducted by the Japan Pancreas Society, demonstrated that the five-year survival rates for stage 0, Ia, and Ib were 85.8%, 68.7%, and 59.7%, respectively, suggesting that early detection of pancreatic cancer can result in a good prognosis [3]. Unfortunately, however, the number of patients with stage 0, Ia, and Ib was 1.7%, 4.1%, and 6.3% of the target cases, respectively, indicating that early diagnosis was not easy. Based on the facts, early detection and systemic treatments should be required to improve the prognosis of as many patients with pancreatic cancer as possible; however, a nationwide large-scale health survey of pancreatic cancer screening has not been conducted.

One of the significant reasons why pancreatic cancer has a poor prognosis is that there is no technique and medical strategy for early detection of pancreatic cancer [4,5]. In particular, pancreatic ductal adenocarcinoma (PDAC) is the most common type of pancreatic cancer with very high mortality [6]. The most effective solution to improve the prognosis of pancreatic cancer is thought to be both early detection and diagnosis, as the prognosis of early-stage pancreatic cancer is significantly more favorable [7]. Nonetheless, early diagnosis is currently challenging due to its non-specific symptoms and anatomical location, along with the lack of reliable early-stage biomarkers [8,9]. In addition, the requirement of more examinations for definitive diagnosis of pancreatic cancer than other diseases may become a barrier to proceed with early detection. Thus, there is a significant need for new diagnostic imaging strategies for the detection of pancreatic cancer in the early state. In this study, we retrospectively investigated pancreatic morphological abnormalities leading to the early diagnosis of PDAC with CT imaging.

## 2. Methods

### 2.1. Study Design and Patient Population

We retrospectively reviewed 308 patients diagnosed with pancreatic cancer by CT in our institution between 2011 and 2017. Key exclusion criteria were the presence of other diseases (67 patients) and no other abdominal CT image in our institution taken before diagnosis of pancreatic cancer (200 patients). In total, 41 out of 308 patients were enrolled in this study (Figure 1). As a control group, 4277 patients with no pancreatobiliary diseases were enrolled.

### 2.2. Study Variables

Data relating to patient characteristics, location of the pancreatic cancer, and abdominal CT findings such as pancreatic morphological abnormalities and pancreatic duct patency were collected.

### 2.3. Diagnosis Management and Imaging Modality

In total, 41 patients were diagnosed as PDAC using high-quality dedicated CT and/or MRI according to Pancreatic Adenocarcinoma, NCCN Clinical Practice Guidelines in Oncology. Additionally, endoscopic ultrasound-fine needle aspiration or pancreatic juice cytology was conducted preoperatively in cases performed by surgery and chemotherapy. In cases of progressive diseases with poor performance status, diagnostic imaging was only performed.

Three types of diagnostic CT were used in this study: Philips IDT 16 from 2004 to 2008, Phillips Brilliance 64 from 2008 to 2017, and TOSHIBA Aquilion ONE 320 from 2015 to 2017. The slice thickness of these CTs was 1–5 mm. A series of CT images from the oldest date to the latest date was reviewed, and the lesion that eventually becomes pancreatic cancer was evaluated. The patency of the main pancreatic duct around the lesion was evaluated. Additionally, based on the date from the oldest CT image and the latest CT image, the period it took to confirm pancreatic cancer was measured.

### 2.4. Ethical Approval

All subjects enrolled in this research gave their informed consent, which, alongside the described protocol, has been approved by the ethic committee of Tokyo Dental Collage Ichikawa General Hospital (#I 18-60).

## 3. Results

### 3.1. Clinical Characteristics

In total, 20 male and 21 female patients, with an average age of 74.8 ± 10.5 years (range: 48–87 years) were included in this study (Table 1). The latest CT revealed 18 patients with pancreatic head cancer, 15 patients with pancreatic body cancer, 5 patients with pancreatic uncinate cancer, and 3 patients with pancreatic tail cancer. Abdominal CT examination was performed four times on average.

### 3.2. A Representative Case and Classification by Specific Abnormality of the Pancreas

An analysis of CT imaging revealed two kinds of specific pancreatic morphological abnormalities: local K-shaped constriction of the pancreatic parenchyma (K-sign, Figure 2) and localized fatty change. For example, a representative case in patients with K-sign is shown in Figure 3. This patient developed pancreatic cancer approximately two years after the appearance of K-sign by CT. Although visible tumor invasion to the pancreatic surface was unascertainable, the cross-section surface of the macroscopic specimen showed that a thickness of pancreas in #9 to #11 was thinner than that in the head-side and tail-side, which matched up clearly with the K-sign of CT (Figure 3). Based on these findings and the definition, all patients were assigned into three groups below.

Group A: K-sign. Two or more consecutive slices of axial CT image showed the K shape and the main pancreatic duct was patent.

Group B: localized fatty change. Axial CT image showed no diffuse fatty change of pancreas, and also their oldest CT showed no sign of the localized fatty changes.

Group C: No abnormality. Axial CT image showed no evidence of pancreatic abnormal lesion.

The results of classification are shown in Table 2. Classification due to CT findings of all patients showed 24 patients in Group A (58.5%), 8 patients in Group B (19.5%), and 9 patients in Group C (22.0%). In Group A, K-sign in the pancreatic body (12 patients, a representative patient shown in Figure 4A), head (9 patients, a representative patient shown in Figure 4B), tail (2 patients, a representative patient shown in Figure 4C), and uncinate (1 patient) was developed. In particular, 9 out of 24 patients in Group A showed no dilatation of the main pancreatic duct around K-sign (Figure 4D). Furthermore, pancreatic cancer with a K-sign in the head and body accounted for 87.5% in Group A. In Group B, localized fatty change in the pancreatic head (4 patients, a representative patient shown in Figure 5A) and uncinate (four patients, a representative patient shown in Figure 5B) were developed. Group C accounted for four patients in pancreatic head (Figure 6), three patients of body, one patient in uncinate, and one patient in tail. All nine patients in Group C showed severe stenosis or complete occlusion of the main pancreatic duct. Out of 4277 control patients, 7 patients (0.16%) showed K-sign.

### 3.3. Time to Development of Pancreatic Cancer

The average time of development of pancreatic cancer in all patients was 21.8 months for Group A (range: 2–62 months), 19.0 months for Group B (range: 8–38 months), and 70.4 months for Group C (range: 34–128 months).

## 4. Discussion

The prognosis of PDAC has not improved despite the progress of diagnostic modality. Both now and in the past, how to detect a small pancreatic tumor seem to be important because one report demonstrated that the stage of cancer progression in 80% of patients with pancreatic tumor less than 2 cm was progressing over stage III [10]. In one well-known study that addressed this issue, periods from pancreatic tumorigenesis until the birth of the founder cell of a parental clone and until the birth of the cell to the lesion were 11.7 and 6.8 years on average, respectively, suggesting that it will take approximately 15 years to detect as a diagnosable tumor [11]. Additionally, only 25% of patients with a diagnosable pancreatic tumor were symptomatic [12], indicating that PDAC is extremely hard to diagnose. On the other hand, there are several risk factors for PDAC such as a family history of pancreatic cancer, hereditary or chronic pancreatitis, intraductal papillary mucinous neoplasm, diabetes, obesity, smoking, and alcohol [13,14]. Screening of PDAC for the high-risk group above has reached a consensus; however, it was further argued that screening of PDAC for the general population should not be performed [15,16]. There is a high possibility of losing the balance of advantages and disadvantages by performing highly invasive examinations on all patients with potential pancreatic cancer. As a feasible examination to the general population, a blood examination for PDAC can be performed. In particular, carbohydrate antigen 19-9 (CA19-9) is superior for PDAC detection because of its high positive sensitivity. When the reference value of CA19-9 was 74 U/mL, the specificity was 100% [17]. In addition, a novel tumor marker, “thrombospondin-2”, has been noted for its high detection rate for PDAC [18]. Yet specific conditions with systemic inflammation and cardiac diseases could induce false positives, and it remains unclear how sensitive tumor markers are. That is to say, the usefulness of tumor markers as an early diagnosis remains questionable. Until now, we could not gain a broad consensus on inspecting uniformly tumor markers with low sensitivity and specificity and performing annual invasive examinations. Therefore, the discovery of specific morphological changes that provide clues for early detection of pancreatic cancer with accurate evaluation of the pancreas using noninvasive imaging modalities may become increasingly important.

For the diagnosis of pancreatic cancer, various modalities such as CT, magnetic resonance imaging, and ultrasonography are commonly performed; however, there have been few studies investigating the predictive findings of pancreatic cancer development by the detection of earlier pancreatic morphological abnormalities. This is thought to be one of the reasons why pancreatic cancer has a poor prognosis. In one study that addresses this issue, the earliest findings on the development of an early-stage pancreatic cancer were pancreatic duct dilatation and cutoff [19]. The CT findings, pancreatic duct dilatation, and cutoff in the study potentially indicates occlusion of the pancreatic duct by pancreatic cancer. Additionally, one report demonstrated that focal hypoattenuation, pancreatic duct dilatation, pancreatic duct interruption, distal parenchymal atrophy, and contour abnormality in pre-diagnostic CT findings of 16 patients accounted for 75%, 50%, 45%, 45%, and 15%, respectively [20]. These two studies did not investigate the morphological or contour abnormality of the pancreas in detail. CT findings in our study showed that pancreatic duct abnormalities such as pancreatic duct dilatation and cutoff were not detected alone and were included in the K-sign, which is consistent with a previous report that found a correlation between the development of pancreatic cancer derived from pancreatic duct epithelium and the morphological change of the pancreatic parenchyma around pancreatic cancer [21]. Among previous studies on pancreatic morphological abnormalities, one report demonstrated the positive correlation between the degree of fatty infiltration in the pancreas and pancreatic ductal adenocarcinoma [22]; however, the evidence within this report is currently fragmentary due to the lack of reliable diagnostic imaging findings. Our CT finding, “K-sign”, a localized morphological change of the pancreas, may have been caused by a combination of the development of pancreatic cancer in the pancreatic duct epithelium and fatty change around the pancreatic parenchyma. The mechanism underlying the K-sign remains unclear, but our study potentially revealed an important insight in terms of presenting more pronounced morphological abnormalities than previous studies and almost no K-sign in patients with no pancreatobiliary diseases (0.16%). While a direct comparison between these previous studies and our study is difficult, our data demonstrating that approximately 60% of patients with pancreatic cancer demonstrated K-sign suggested that these signs could provide a new early diagnostic finding of pancreatic cancer development. Moreover, although these CT findings do not necessarily reflect a direct correlation with pancreatic cancer development, our prospective observation of patients with K-sign revealed that the existence of K-sign could facilitate early diagnosis of pancreatic cancer development.

The other significant finding shown in this study is the observation of an approximate period from the appearance of K-sign to the development of pancreatic cancer. Our data demonstrates that patients with K-sign and localized fatty change developed pancreatic cancer within a few years, which may provide important insights regarding the timeline for the development of pancreatic cancer (Group A, 21.8 months; Group B, 19.0 months). Nonetheless, it is important to note that patients in Group C took a longer time to develop pancreatic cancer that those in other groups (19.0–21.8 versus 34–128 months). Although our current study did not reveal the existence of pancreatic morphological abnormalities in Group C, it is clear that patients in Group C showed K-sign in cases where they had undergone abdominal CT long before developing pancreatic cancer. A prospective study investigating the correlation between K-sign and the time to onset is currently in progress, which may facilitate more effective and precise diagnosis and consequently induce a better prognosis.

Our study on imaging diagnosis before development of pancreatic cancer required the long and precise follow-up of pancreatic morphologic abnormalities such as K-sign. For the diagnosis of pancreatic cancer, endoscopic ultrasonography (EUS) is commonly used. Nonetheless, while EUS could detect 93.5% of the stenosis of the main pancreatic duct in patients with pancreatic carcinoma in site, EUS could only detect 56.3% of the hypoechoic area shown as early-stage pancreatic cancer [23]. CT findings “K-sign and fat changes,” which are hard to detect with EUS, may help in the accurate diagnosis after the appearance of pancreatic cancer. Generally, CT imaging has some shooting conditions: non-contrast CT, single helical CT, and multi-detector row CT (MDCT). In particular, several studies have reported that MDCT could be useful for the observation and evaluation of the pancreas by optimal pancreatic parenchymal and peripancreatic vascular enhancement [24,25,26]. Additionally, MDCT facilitates multiplanar reconstructions such as curved planar reformations, which led to precise detection of early-stage pancreatic cancer. However, the sensitivity of pancreatic cancers smaller than 10 mm by MDCT is relatively lower. More accurate diagnostic imaging appears to be required to identify the development of an early-stage pancreatic cancer itself. Although there may be challenged associated with early-stage pancreatic cancer in facilities with various degrees of imaging modalities, the K-sign detected in this study has the added advantage as an early diagnostic indicator in that it can be detected in all parts of the pancreas under shoot conditions of abdominal CT such as contrast and non-contrast CT and MDCT. Therefore, K-sign detected on any CT is potentially a meaningful diagnostic imaging finding suggestive before the appearance of pancreatic cancer.

## 5. Conclusions

The K-sign is an important CT finding as a pancreatic morphological abnormality that may be used as a reliable predictor of pancreatic cancer development.

## 6. Limitations

This present study was subject to some limitations. It is a retrospective study, and abdominal CT in this study was not performed under the same conditions and follow-up period. Additionally, the localized constriction of the pancreas was very hard to quantify because the K-sign is one of the image findings. Subsequent prospective studies on the exact timing of pancreatic cancer development in patients with K-sign will provide a further rationale.

## Figures and Tables

**Figure 1 cancers-13-04222-f001:**
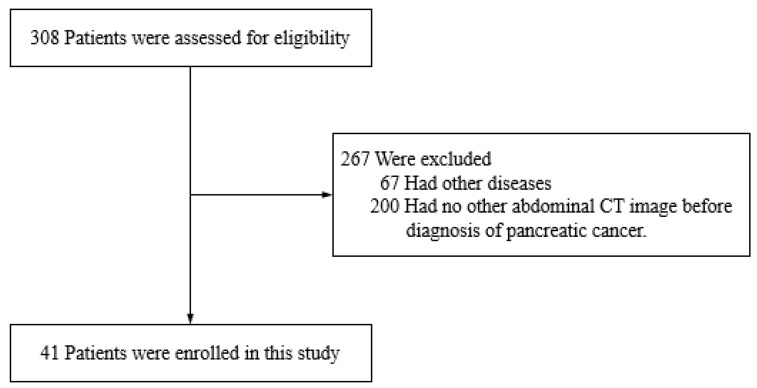
Enrollment.

**Figure 2 cancers-13-04222-f002:**
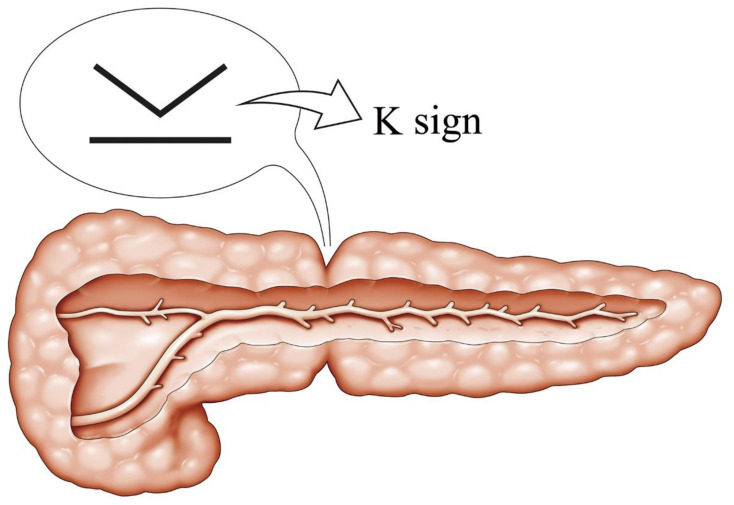
Illustration and definition of K-sign. K-sign means a localized constriction of the pancreatic parenchyma. It looks “K” on axial CT image.

**Figure 3 cancers-13-04222-f003:**
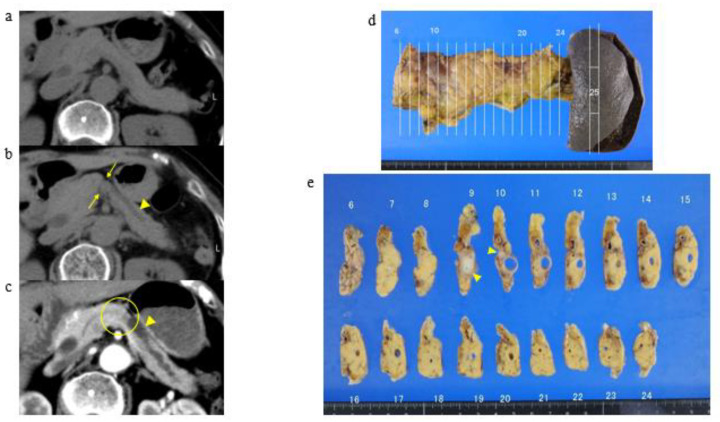
A representative case with K-sign (pStage IIB). (**a**) October 2008: pancreatic body shows no evidence of pancreatic abnormality. (**b**) June 2014: K-sign (yellow arrows) and the main pancreatic duct dilatation (an arrowhead) are shown in the pancreatic body. (**c**) August 2016: pancreatic body cancer with main pancreatic duct dilatation is shown in the K-sign area (a yellow circle and arrow head). (**d**) Pancreatosplenectomy was performed in September 2016. Macroscopic specimen did not show visible tumor invasion to the pancreatic surface and proof of K-sign. (**e**) The cross-section surface of the macroscopic specimen showed pancreatic cancer around the main pancreatic duct (yellow arrows).

**Figure 4 cancers-13-04222-f004:**
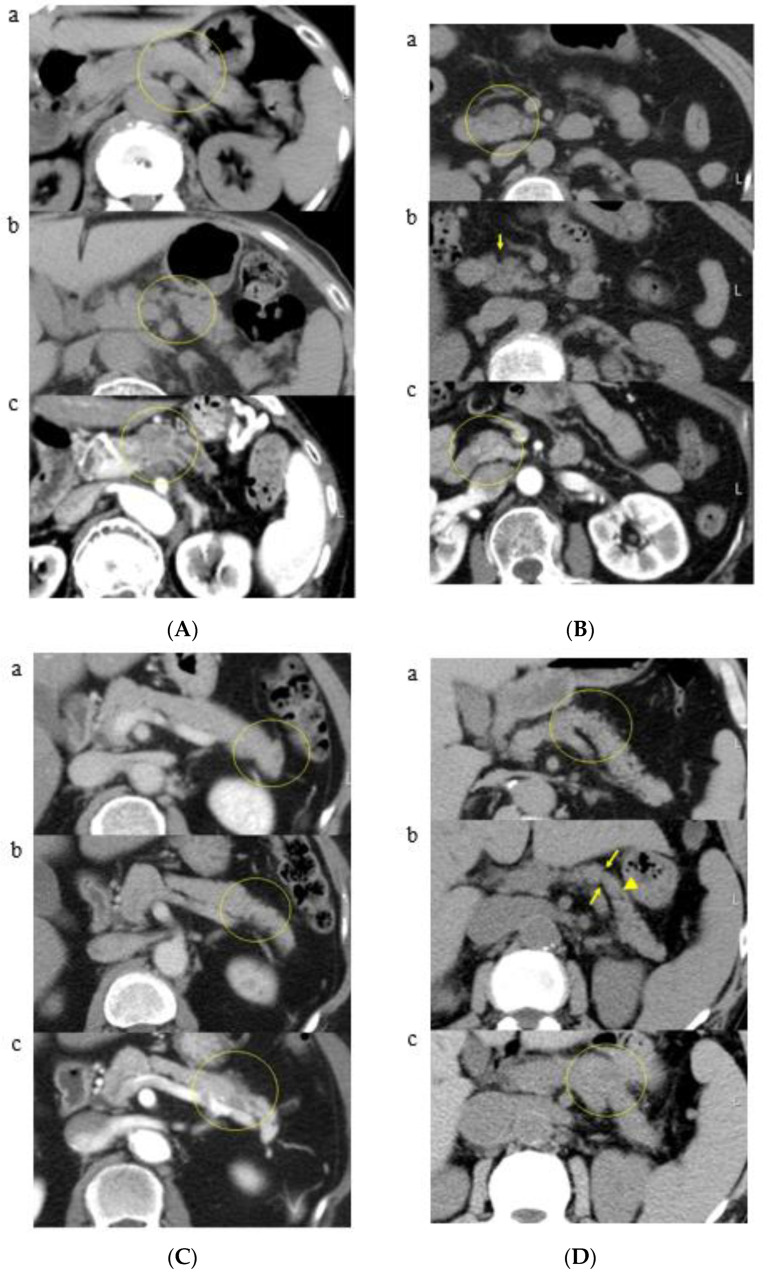
(**A**) Group A (a representative patient with K-sign in the pancreatic body): (**a**) April 2005: pancreatic body shows no evidence of pancreatic abnormality (a yellow circle); (**b**) January 2013: K-sign is shown in the pancreatic body (a yellow circle); (**c**) December 2016: pancreatic body cancer with main pancreatic duct dilatation is shown in the K-sign area (a yellow circle). (**B**) Group A (a representative patient with K-sign in the pancreatic head): (**a**) May 2004: pancreatic head looks normal on this CT (a yellow circle); (**b**) September 2012: K-sign is shown in the pancreatic head (a yellow arrow), (**c**) July 2013: pancreatic head cancer is developed at the site of K-sign (a yellow circle). (**C**) Group A (a representative patient with K-sign in the pancreatic tail): (**a**) October 2007: K-sign is shown in the pancreatic tail (a yellow circle); (**b**) October 2011: K-sign expanded compared with the previous CT in 2007 (a yellow circle); (**c**) August 2012: pancreatic tail cancer with main pancreatic duct dilatation is shown (a yellow circle). (**D**) Group A (a representative patient with K-sign and patented main pancreatic duct): (**a**) March 2003: pancreatic body looks normal on this CT (a yellow circle); (**b**) August 2011: K-sign (a yellow arrow) and main pancreatic duct (a yellow triangle) are shown; (**c**) August 2013: pancreatic body cancer is shown at the site of K-sign (a yellow circle).

**Figure 5 cancers-13-04222-f005:**
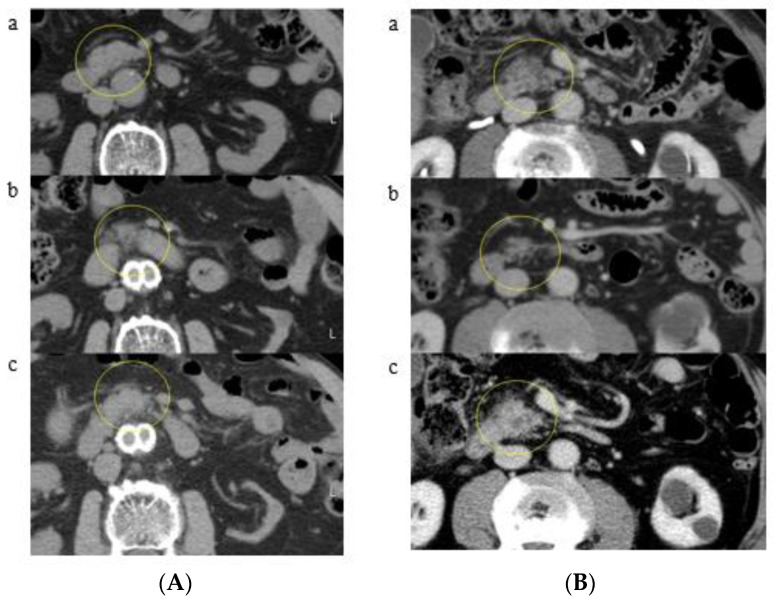
(**A**) Group B (a representative patient with localized fatty change in the pancreatic head): (**a**) July 2006: pancreatic head looks normal on this CT (a yellow circle); (**b**) April 2013: localized fatty change is shown in the pancreatic head (a yellow circle); (**c**) October 2014: pancreatic head cancer is shown at the site of localized fatty change (a yellow circle). (**B**) Group B (a representative patient with localized fatty change in the pancreatic uncinate): (**a**) August 2014: localized fatty change is shown in the pancreatic uncinate (a yellow circle); (**b**) July 2016: localized fatty change expanded compared with the previous CT in 2014 (a yellow circle); (**c**) January 2017: pancreatic uncinate cancer is shown at the site of localized fatty change (a yellow circle).

**Figure 6 cancers-13-04222-f006:**
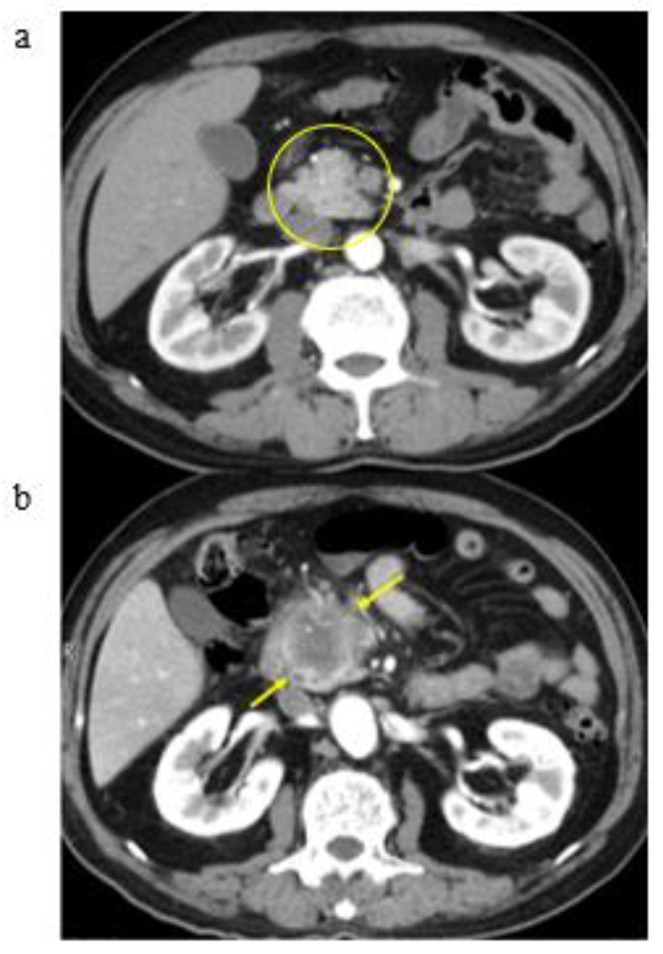
Group C (a representative patient with no abnormality): (**a**) April 2009: pancreatic head looks normal with well enhancement (a yellow circle); (**b**) December 2017: pancreatic head cancer is shown (yellow arrows).

**Table 1 cancers-13-04222-t001:** Patient characteristics.

Characteristic	*n* = 41
Age, y, mean (±SD)	74.8 (±10.5)
Female gender, *n* (%)	21 (51.2)
Location of the pancreatic tumor, *n* (%)	
Pancreatic head	18 (43.9)
Pancreatic body	15 (36.6)
Pancreatic uncinate	5 (12.2)
Pancreatic tail	3 (7.3)

**Table 2 cancers-13-04222-t002:** Classification of CT findings.

Group	CT Finding	*n* = 41 (%)	Location of Pancreatic Cancer (%)			
			Head	Body	Uncinate	Tail
Group A	K-sign	15 (36.6)	5 (33.3)	7 (46.7)	1 (6.7)	2 (13.3)
	K-sign + pancreatic duct patency	9 (21.9)	4 (44.4)	5 (55.6)	0 (0)	0 (0)
Group B	Localized fatty change	7 (17.0)	4 (57.1)	0 (0)	3 (42.9)	0 (0)
	Localized fatty change + pancreatic duct patency	1 (2.4)	0 (0)	0 (0)	1 (100)	0 (0)
Group C	No abnormality	9 (21.9)	4 (44.4)	3 (33.3)	1 (11.1)	1 (11.1)

## Data Availability

The data used to support the findings of this study are available from the corresponding author on request.

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
