# Peer review of "The “K-Sign”—A Novel CT Finding Suggestive before the Appearance of Pancreatic Cancer"

_cancers, 2021, doi:10.3390/cancers13164222_

Round 1
Reviewer 1 Report
This manuscript reported by Kobashi et al. is well documented about "K-sign" for diagnosis of early PDAC. However, there are many problems to be dissolved in the current status.
- The number of pancreatic cancer (PC) is too small. The author should enroll more than 200 cases of PC from multiple institutions to discuss the "K-sigh".
- The title of this article does not match the content. Recently, the case of stage 0 or I (UICC) pancreatic cancer should be considered to be an early-stage case. The author should evaluate the CT image findings of stage 0 and I PC cases deeply.
- The author should demonstrate how to diagnose each case as PC histologically. What kinds of methods did they perform, EUS-FNA or surgery?
- The author did not demonstrate any data of clinical stages, nor pathological stages of PC.
Author Response
Comments to the Author
- The number of pancreatic cancer (PC) is too small. The author should enroll more than 200 cases of PC from multiple institutions to discuss the "K-sigh".
Reply: Thank you for your reminder. The initial study design included 308 patients with PDAC, but 41 patients out of 308 patients were enrolled in this study according to the exclusion criteria shown in (new) Figure 1. In addition, based on the insights of this study, in April 2019 we started a prospective study of patients who showed K-sign in CT scan by chance.
- The title of this article does not match the content. Recently, the case of stage 0 or I (UICC) pancreatic cancer should be considered to be an early-stage case. The author should evaluate the CT image findings of stage 0 and I PC cases deeply.
Reply: Thank you for the suggestion. Indeed, many cases with pancreatic cancer at the stage 0 and 1 should be investigated. We adopted the title not because K-sign was found in the patients with advanced pancreatic cancers, but because K-sign was early found in a retrospective study of patients with advanced pancreatic cancer. Again, we started a prospective study of patients who showed K-sign in CT scan by chance. Unfortunately, the latest results of a prospective study cannot be shown in this study, but we would appreciate your understanding.
- The author should demonstrate how to diagnose each case as PC histologically. What kinds of methods did they perform, EUS-FNA or surgery?
Reply: 41 patients were diagnosed as PDAC by high-quality dedicated CT and/or MRI according to Pancreatic Adenocarcinoma, NCCN Clinical Practice Guidelines in Oncology. Additionally, EUS-FNA or pancreatic juice cytology was conducted preoperatively in cases performed by surgery and chemotherapy. In cases of progressive diseases with poor performance status, diagnostic imaging was only performed. We added the above description in the Methods.
- The author did not demonstrate any data of clinical stages, nor pathological stages of PC.
Reply: Thank you for your suggestion. This study accounted for the majority of UICC clinical stage 3 or 4. We focus on not the clinical stage at the time of diagnosis with pancreatic cancer but K-sign as a CT finding before diagnosis with pancreatic cancer. In addition, we did not describe the clinical and pathological stage because all patients but a patient with stage 2B shown in new Fig 3 died of the progression after diagnosis of pancreatic cancer.

Reviewer 2 Report
In the current study, Kobashi, et al aimed to explore the novel indicator of early pancreatic cancer on abdominal CT imaging. They focused on the pancreatic morphological abnormalities, and examined “K-sign” and localized fatty change detected in CT imaging. Finally, they concluded that the K-sign could be an important CT finding as a pancreatic morphological abnormality, and might be used as a reliable predictor of early-stage pancreatic cancer development. This topic, that is, early diagnosis and detection of pancreatic cancer is recent one of the most important assignments to improve the dismal survival outcomes of pancreatic cancer. Therefore, this study is not only clinically necessary but also extremely original. I think that this paper will be acceptable with some minor revision as described below.
- The authors examined the CT findings regarding the K-sign, and compared with those in 4,277 control patients. As a result, the K-sign was observed in 0.16% of control patients as well. What does it mean? Is the K-sign not specific? Please add the comment in the discussion section.
- In Table 1B, the K-sign was observed in every location of pancreatic cancer. Is there any difference or location specific feature of this sigh? Please add the comment.
- The data regarding average times to development of pancreatic cancer is a little bit unclear. Please define this parameter in details, and add the description.
Author Response
Comments to the Author
- The authors examined the CT findings regarding the K-sign, and compared with those in 4,277 control patients. As a result, the K-sign was observed in 0.16% of control patients as well. What does it mean? Is the K-sign not specific? Please add the comment in the discussion section.
Reply: 58.5% of patients with pancreatic cancer had a K-sign. At the timing of the initial study design, it was not clear whether K--sign was an absolute CT finding of the pancreatic cancer diagnosis. So, 4277 patients with no pancreatobiliary diseases as a control group were enrolled, and seven patients (0.16%) showed K-sign. Although a direct comparison cannot be made, K-sign is likely to be a promising CT finding from 58.5% versus 0.17%.
- In Table 1B, the K-sign was observed in every location of pancreatic cancer. Is there any difference or location specific feature of this sigh? Please add the comment.
Reply: Thank you for your suggestion. It was difficult to make statistically significant differences among the locations of pancreatic cancer. As you pointed out, however, the degree of the observation in each location was likely to be different. In this study, we experienced that body and tail were easy to be observed and uncinate was the most difficult to be done. In April 2019, we started a prospective study of patients who showed K-sign in CT scan by chance. The trial may reveal the features in each location.
- The data regarding average times to development of pancreatic cancer is a little bit unclear. Please define this parameter in details, and add the description.
Reply: Thank you for the suggestion. Regarding Time to development of pancreatic cancer, we collected a series of CT images from the oldest date to the latest date in all 41 patients, and evaluated the lesion that will eventually become pancreatic cancer and the patency of the main pancreatic duct around the lesion.

Reviewer 3 Report
Thank you for the opportunity to review this paper. This is a single-center retrospective study to investigate the diagnostic impact of CT findings in patients with PDAC. Authors newly defined “K-sign” as a sign of early PDAC, and concluded that “K-sign” may be used for the early diagnosis of PDAC by CT imaging. This article contained important information and can resolve clinical question for early diagnosis of PDAC. Then, the followings are the list of my comments which should be addressed and can increase the value of this manuscript.
Comments
- In the abstract, author said that 41 patients were diagnosed with pancreatic cancer between 2011 and 2017. However, in the methods section, the number of the patients with PDAC was 308 patients. Please correct the description.
- To avoid misunderstanding, please add patients flow chart of the study population as figure 1.
- The inclusion and exclusion criteria should be described in detail.
- Results of the control group should be mentioned in the results section. What is the diagnosis of 7 patients who had K-sign in the control group?
- K-sign can be used even in no-enhanced CT? Or, only in enhanced CT? Describe the author's ideas shortly in the discussion section.
Author Response
Comments to the Author
- In the abstract, author said that 41 patients were diagnosed with pancreatic cancer between 2011 and 2017. However, in the methods section, the number of the patients with PDAC was 308 patients. Please correct the description.
Reply: Thank you for your reminder. The initial study design included 308 patients with PDAC, but 41 patients out of 308 patients were enrolled in this study according to the exclusion criteria shown in (new) Figure 1.
- To avoid misunderstanding, please add patients flow chart of the study population as figure 1.
Reply: Thank you for your suggestion. We added patient enrollment as Fig 1.
- The inclusion and exclusion criteria should be described in detail.
Reply: Thank you for your suggestion. We added patient enrollment as Fig 1.
- Results of the control group should be mentioned in the results section. What is the diagnosis of 7 patients who had K-sign in the control group?
Reply: Thank you for your suggestion. We added the result of the control group. At the timing of the initial study design, it was not clear whether K--sign was an absolute CT finding of the pancreatic cancer diagnosis. So, 4277 patients with no pancreatobiliary diseases as a control group were enrolled, and seven patients (0.16%) showed K-sign. Although a direct comparison cannot be made, K-sign is likely to be a promising CT finding from 58.5% versus 0.17%. Unfortunately, we couldn't follow up on these 7 patients, so we can't show if they had pancreatic cancer.
- K-sign can be used even in no-enhanced CT? Or, only in enhanced CT? Describe the author's ideas shortly in the discussion section.
Reply: Thank you for your reminder. This study included both conditions of abdominal no-enhanced and enhanced CT. we added the comment in Discussion section.

Round 2
Reviewer 1 Report
This manuscript is well revised according to reviewer's comments. However, the number of early pancreatic cancer (PC) cases is too small. The title of this manuscript contains "early diagnosis", therefore, more cases diagnosed at an early-stage including Stage IA or O PC must be evaluated from multiple institutions.
Author Response
2nd Comments to the Author
- This manuscript is well revised according to reviewer's comments. However, the number of early pancreatic cancer (PC) cases is too small. The title of this manuscript contains "early diagnosis", therefore, more cases diagnosed at an early-stage including Stage IA or O PC must be evaluated from multiple institutions.
Reply: Thank you for your reminder. Indeed, our study has only a few patients with early PC. So, as you pointed out, our title as it is can be misleading. We changed our title into The “K-sign” - A novel CT finding suggestive before the appearance of pancreatic cancer. Additionally, we deleted the descriptions that we could diagnose "early" PC at the "early" stage. We would appreciate your understanding.

Round 3
Reviewer 1 Report
This manuscript reported by Kobashi et al. is well revised according to reviewer's comment. It should be considered to be accepted to "Cancers" after minor spell checked.